# Broccoli-Soybean-Mangrove Food Bar as an Emergency Food for Older People during Natural Disaster

**DOI:** 10.3390/ijerph18073686

**Published:** 2021-04-01

**Authors:** Fatmah Fatmah, Suyud Warno Utomo, Fatma Lestari

**Affiliations:** 1Disaster Management Study Program, School of Environmental Science, Universitas Indonesia, Jakarta 10430, DKI Jakarta Province, Indonesia; 2Department of Environmental Health, Faculty of Public Health, Universitas Indonesia, Depok 16424, West Java Province, Indonesia; suyudwarno@gmail.com; 3Department of Occupational Health and Safety, Faculty of Public Health, Universitas Indonesia, Depok 16424, West Java Province, Indonesia; fatma@ui.ac.id

**Keywords:** natural disaster, older people, weight, broccoli-soybean-mangrove food bar, food emergency product

## Abstract

Older people risk poor nutritional status during natural disasters due to low intakes of energy, carbohydrates, protein, and fat. A food bar is a form of emergency food product that contains carbohydrate and protein, and is practical for disaster situations. The study aimed to investigate the effect of a broccoli-soybean-mangrove food bar on older people’s weight following natural disasters. A quasi-experimental pre-post intervention study was designed using 33 subjects at the treatment group of various nutritional status types of older people during two weeks with balanced nutrition education over two weeks. Bivariate analysis with a paired *t*-test used to test whether weight, macronutrient intakes, and balanced nutrition knowledge were significantly different before and after the study. The study showed broccoli-soybean-mangrove food bar consumption resulted in a significantly increased weight of 0.2 kg, energy (291.9 kcal), protein (6.1 g), carbohydrate (31.1 g), dan fat (15.6 g) intakes. Balanced nutrition education of older people could also substantially increase knowledge of older people regarding nutrition (11.8 points). The proportion of malnourished subjects who gained weight was more remarkable than normal subjects in the first and second weeks of the intervention. However, the proportion of normal nutritional status subjects having increased macronutrients intakes was higher than the malnourished subjects. These findings recommend broccoli-soybean-mangrove food bar consumption to significantly improve weight and macronutrients intakes in older people following a natural disaster. It is necessary to make the broccoli-soybean-mangrove food bar more available, accessible, and affordable to all people in emergencies, mainly for older people.

## 1. Introduction

Natural disasters in Indonesia, such as floods, landslides, and earthquakes, continue to increase from year to year. The intensity of natural disasters in 2019 increased by 32.4% (1107 cases) compared to 2018 with 836 cases [1]. Hydro-meteorological disasters are the most frequent type in Indonesia [2]. One of the natural disasters that claimed many older people as casualties was the tsunami earthquake in Aceh on 26 December 2004 [3]. One of the main problems faced in disaster management is how to meet the need for food in disaster-prone groups, which also include older people, due to the challenge to meet nutritional needs in a disaster context. Other risk factors for older people’s vulnerability in disasters are reduced access to food, increased degenerative diseases and infections, psychosocial stress, and poverty. Older people in the affected areas should have easy access to food that is easy to prepare. The food for older people in a disaster situation should also meet this group’s protein and micronutrient requirements [4].

The older people in disaster areas often experience malnutrition due to low energy intake, carbohydrates, protein, and fat. While the older people require small portions of food, the food must be nutrient-dense and easy to digest [5,6,7]. Moreover, factors supporting the incidence of malnourished older people are decreased appetite, decreased ability of the body to digest food, limited power to buy and prepare food, decreased health status, psychological factors from isolation and depression, which in turn affect the nutrition status of older people. One means of improving older people’s malnutrition status in disaster areas is supplementing energy-dense snack foods [8]. Several nutrition-dense food intervention studies for malnourished older people in the form of snacks, meals, and home delivery meals have been shown to improve their nutritional status [9,10,11,12].

With that in mind, the need for developing emergency food products (EFPs) is apparent. The EFP is a food product designed to be used in emergencies, be consumed directly, and meet daily nutritional needs. The purpose of giving EFP is to provide nutritious food to meet the daily recommended dietary allowance (RDA) for 15 days, starting from the recognized time of displacement [13]. Food bars are appropriate for emergency food in disaster situations because they are made from a mixture of local food ingredients in a stable, compact form with high carbohydrates and protein contents. It is also practical as it can be consumed immediately without the need for further processes. Food bars are more resistant to pressure than biscuits/cookies because they are semi-wet food products [14]. Some other advantages of the food bar as an emergency food product for older people victims in natural disasters include among others, that it is easy and quick to distribute, requires no cooking equipment, is durable, and is suitable as a temporary food supplement [15].

Food bars are one of the emergency food products produced using local raw materials that are easy to obtain but with a higher economic value due to its special design. The food bar in this study was mainly made from mangrove fruit/*lindur* fruit flour (*Bruguiera gymnorrhiza*) and mixed with soy flour and broccoli flour. Mangrove fruit is the fruit of a type of mangrove plants commonly found in Indonesian waters. This flour form was selected because this fruit has a high carbohydrate content and can be processed into flour. The texture of mangrove fruit flour was similar to the texture of wheat flour, but the color was darker. Soybean and broccoli flours were also used as raw materials for the food bar because these are local foods used to support food security in Indonesia [16].

A study on the development of an emergency food product in the form of a food bar that is made from a mixture of mangrove fruit or *lindur* fruit flour (*Bruguiera gymnorrhiza*) (100%), broccoli flour (50%), and soy flour (50%) was performed in August 2020 in Depok City, West Java Province, Indonesia. The study objective was to assess the effect of food supplementation in the form of a food bar made of a mixture of mangrove flour, flour broccoli, and soybean flour on the weight of older people following a natural disaster. The three essential ingredients were selected for the following reasons. The mangrove fruit has a high carbohydrate content and can be used as a substitute for wheat flour in making biscuits [17]. Broccoli contains lots of vitamin C (16 mg), iron (3.4 mg), and zinc (0.6 mg). At the same time, soybeans are high in calories (189 kal), zinc (3.7 mg), iron (3.9 mg), and calcium (91 mg) needed by the body to increase immunity [18]. By consuming 50 g of the broccoli-soybean-mangrove food bar in one day (246 calories), the needs for 14–16% of male and female older people of 1800 calories and 1550 calories, respectively, will be fulfilled [19].

## 2. Materials and Methods

### 2.1. Study Design

The pre experimental study design was applied to 33 subjects in the treatment group without a control group (the one group pre-post test). Ethical clearance was obtained from the Health Research Ethics Commission of the Faculty of Public Health, Universitas Indonesia (No.:293/UN2.F10.D11/PPM.00.02/2020). Subjects signed a written informed consent before the study began in early August 2020.

### 2.2. Population and Sample

The treatment group was given broccoli-soybean-mangrove food bars during two weeks. The reason why the duration of intervention was only two weeks is because emergency food is provided for refugees or victims of natural disasters for 15 days from the time a disaster is experienced [20]. The inclusion criteria for the subjects were 60 years or above; male and female; having different types of nutritional status (underweight, normal, and overweight) because emergency food is not suitable for malnourished individuals who require medical care [21]; and victims of flooding in early January 2020 in one of the two selected urban villages (*Kelurahan*). The study sites comprised Cilodong and Kalibaru villages in Depok City. The two locations were selected based on the largest number of victims during the floods in January 2020. The exclusion criteria were: suffering from certain chronic diseases and illiteracy. The calculation of the minimum sample size for this study follows the hypothesis test of paired mean difference [22]. The subjects gained 0.5 kg in weight after they obtained meals enriched the dense food during 12 weeks. We examined the standard deviation of paired data from the previous study = 1.5 [23]; a two-sided significance level of 0.05, a sample size of *n* = 20 is needed to reach 90% statistical power.

### 2.3. Subject Recruitment

The number of subjects who met the inclusion criteria until the end of the study was 33, which is less than the minimum required sample size of 40 subjects (Figure 1). At the start of the study, the total number of subjects involved was 40 subjects in the treatment group. However, two subjects withdrew from participation because they thought that the broccoli-soybean-mangrove food bar was too sweet for them and they became bored of eating it. Another 5 subjects withdrew from the control group because of boredom, itching, and swollen extremities after consuming the food bar in week 1. One subject was seriously ill so she had to be admitted to hospital.

### 2.4. Instrument

Height and mid-upper arm circumference (MUAC) were measured at the start of the study. Weight was measured two times (in the morning and afternoon) at week one and week two. A digital scale with 0.1 kg precision was used for weighing, and a microtoise with an accuracy of 0.1 cm was used for measuring height. The midline band was used to measure the mid-upper arm circumference (MUAC). The level of independence of the subjects was assessed using BADL (Basic Activity Daily Living) and IADL (Instrument Activity Daily Living) tools [24]. In contrast, the risk of dementia or possible cognitive impairment was assessed using the MMSE (Mini-Mental State Examination) tool [25]. The Mini Nutritional Assessment (MNA) was used to determine the risk of malnutrition in the subjects [26]. The data on the risk of malnutrition, level of independence, and cognitive impairment were collected at the start of the study.

### 2.5. Food Bar Cooking Process

The broccoli-soybean-mangrove food bar was made from a mixture of melted butter, chicken egg, refined white sugar, wheat flour, mangrove flour, broccoli flour, soy flour, and chocolate pasta. A mixer, digital oven, cookie-cutter, tray, digital food scale, and rolling pin were used for cooking the food bar. The food bar was prepared as follows:Beat melted butter and white sugar with a mixer until smooth/homogenous. Put all the eggs and the mangrove, broccoli, and soybean flour into the dough.Stir the dough with the wish tool until blended and put the chocolate paste or other flavor into the dough. Then, stir homogenously with a wish or spatula tool.Prepare a square baking sheet with white parchment paper, then spread with butter until blended.Pour the dough into the pan and baked it in the oven for 30 min at a temperature of 180 °C. Continue baking again at 180 °C for 20 min to get a really dry quality food bar. One of the criteria for a food bar is that it must be completely dry for long-term storage as an emergency food [13].

### 2.6. Nutrition Intervention and Follow-Up

Broccoli-soybean-mangrove food bar supplementation was given to the subjects for two weeks in August 2020. Each subject received 50 g of food bar as a snack during two weeks. Every 50 g of broccoli-soybean-mangrove food bar contained 246 kcal of energy, 30.2 g of carbohydrates, 1.9 g of protein, and 12.1 g of fat [27]. Food bars were made using a mixer, digital oven, cookie-cutter, tray, digital food scale, and rolling pin. The baseline data included demographic characteristics (marital status, age, latest education, last job, and status of living with other people at home), knowledge of balanced nutrition and natural disaster preparedness of older people and were collected before and after the study (at week two). A questionnaire was used for the data collection before and after the intervention to assess the extent of changes in subject knowledge about balanced nutrition in older people and knowledge and practices of older people’s preparedness in facing flood disasters. The daily food consumption data were collected from the subjects using the 24-h food recall form at the start, in the middle, and at the end of the two weeks (a total of 6 days of daily food records for each subject). A weekly food bar distribution form was used to record the number of food bars distributed, the number of food bar consumed, and the remaining food bar in the subject’s home. Education on balanced nutrition for older people and older people’s preparedness to face disasters was performed in the first week using leaflets, flipcharts, and videos developed by the research team.

The organoleptic test comprised the preference test and the hedonic quality test. It was performed on 27 older people who were not included as study subjects (60 years old or above, male and female) as semi trained panelists. In the case of semi-trained panelists, 10–20 people are preferable [28]. All semi-trained panelists did not include the research subjects to avoid bias. They already knew the food bar’s quality (shape, taste, odor, aroma, color) and this can affect their liking, acceptance, and compliance of consuming the food bar. The purpose of the preference and hedonic quality tests were to assess the acceptability of the broccoli-soybean-mangrove food bar among older people, including the preference for aroma, texture, taste, and color. In addition, the test also aimed to assess the fragrance of the aroma, crispness of texture, level of sweetness, and color appeal level. The adherence to food bar consumption in the group was observed by the research team through home visits, which were done three times a week. The purpose of home visits was to record daily food consumption for three days per week, including at the start, in the middle, and at the end of the week, using the 24-h food recall instrument and food bar distribution form. Another purpose of these visits was to record complaints or side effects of the food bar consumption and to assess the subject’s health condition. During these visits, the weight was also measured by taking the older people for weighing at the *posbindu* (older people’s health service post).

### 2.7. Data Analysis

The univariate analysis was performed using SPSS version 23 (IBM, Armonk, NY, USA). It included the frequency distribution of demographic characteristics (age, sex, marital status, latest education level, last job, and the status of living with family), balanced nutrition knowledge of older people. The descriptions of all variables of the study are presented in the tables and figures. The overview of the MMSE, MNA, BADL, IADL, and MUAC is also presented in a similar form. Data on daily food consumption were used to assess macronutrient intakes (energy, carbohydrates, protein, and fat) using the Nutri-survey Program. A bivariate analysis with paired *t*-test was performed to assess changes in the mean weight, macronutrient intakes, and balanced nutrition and preparedness facing natural disaster knowledge of older people at pre-post study. All tests were two-sided, and a *p*-value < 0.01 was considered statistically significant.

## 3. Results

### 3.1. Hedonic and Hedonic Quality Test

The study assessed the preference for the food bar using two tests: the acceptability test and hedonic quality test (Table 1). These tests were performed on 27 semi-trained panelists aged 60 years or above. The hedonic food bar test included an assessment of preferences that included the assessment of aroma, taste, color, and texture. The hedonic quality test assessed the level of fragrance, sweetness, attractiveness, and crunchiness. The majority of the panelists preferred the taste, color, and texture of the broccoli-soybean-mangrove food bar, except the aroma. When asked for an assessment of the hedonic quality of the food bar, most panelists stated that the aroma of broccoli-soybean-mangrove food bar (Figure 2) was a little nice and good (39.3% respectively). Most subjects liked the texture of the broccoli-soybean-mangrove food bar was crunchy, the taste was sweet, and the color was attractive (Table 2).

### 3.2. Sociodemography Characteristic and Nutritional Status, Independence Level, and Cognitive Profiles

Table 3 shows the majority of the subjects were female and widow/widower/divorced with the age range of 71 to 80 years old. More than three-quarters of the total subjects had a low-level educational background. Most of the subjects lived with their children, grandchildren, and children-in-law at their house. At the start of the study, the mean body mass index (BMI) was of normal nutritional status and the proportion of underweight subjects was almost equal to normal nutritional status subjects. The risk of chronic energy deficiency (CED) in the older people was defined as a MUAC of <23 cm. For the risk of malnutrition, as measured using the Mini-Nutritional Assessment (MNA), it was demonstrated that more than three-quarters of the total subjects was at risk of malnutrition/undernutrition. In terms of the dementia risk as assessed using the MMSE tool, it was found that the proportion of subjects experiencing possibility of cognitive impairment was quite high (33.3%). The level of independence of subjects in carrying out daily physical activities, measured by BADL, was mostly classified as independent (90.9%). Likewise, the majority of subjects performed physical activities outside home (IADL) at the independent level (84.8%).

### 3.3. Weight Change, Dietary Intakes, and Natural Disaster Preparedness Knowledge 

#### 3.3.1. Weight Changes and Dietary Intakes

The underweight subjects had a slightly greater proportion of weight gain than the normal subjects in the first and second week of study. The average weight gain of underweight and normal weight subjects in the first week was 0.3 kg, the lowest 0.1 kg, and the highest 0.9 kg. Meanwhile, in the second week of study, the average of both types of nutritional status of subjects was 0.5 kg, with the lowest of 0.1 kg and the highest 1.2 kg (Table 4).

Table 5 shows that the normal nutritional status subjects had an increased intake of energy, protein, fat, and carbohydrates were higher than malnourished subjects. More malnourished subjects experienced a decrease in these four types of nutrient. Mean weight change was significantly different from the baseline to week 2 of the study as shown in Table 6. Of those who gained weight, mean weight increased 0.2 kg. Macronutrient intakes such as energy, protein, fat, and carbohydrate increased significantly in the intervention subjects (*p* < 0.001), indicating that the broccoli-soybean-mangrove food bar was consumed. Table 7 indicates that at the end of the intervention, male subjects had mean energy, protein, fat, and carbohydrate intakes lower than the Indonesian RDA for male and female. Mean protein consumption on male and female subjects were half of that recommended for males and females. Mean intakes of energy and carbohydrate of male and female subjects were three-fifths of the Indonesian RDA, except for fat intake (almost 100%).

#### 3.3.2. Natural Disaster Preparedness Knowledge

Knowledge of balanced nutrition and guidance on preparedness of older people facing natural disasters was given to the subjects once for 30 min in week 2 of the study. The aim of the education was to increase balanced nutrition knowledge for older people in order to improve consuming nutritious foods, including broccoli-soybean-mangrove food bars. Moreover, to increase the knowledge of older people’s preparedness for natural disasters. Balanced nutrition education materials for older people consisted of balanced nutrition guidelines for older people, factors affecting malnutrition in older people and its impact/effect, and the efforts to prevent malnutrition in older people. Disaster definition, types of disasters, disaster impacts, and preparedness efforts for older people to face natural disasters were topics discussed in the education of the older people for facing natural disasters. Both types of education were provided in direct counseling using leaflets, flipcharts, and video playback.

Table 6 shows the increase knowledge of nutrition and natural disasters of older people by 10.9 points. There was a significant difference in the subjects’ knowledge at the end of the study (*p* = 0.009). Of all subjects who experienced flooding, half of the total subjects in both groups claimed that they were usually prepared to deal with natural disasters that may happen any time. Most of them performed self-evacuation to a relative’s house or other people’s house who was not affected by the flood. Most subjects expressed the need for the elderly to prepare for dealing with natural disasters to be able to save themselves. The most widely accessed sources of information on natural disasters were television, radio, and newspapers, in addition to the information from neighbors and neighborhood (*RT*) administrators. More than three-quarters of the subjects stated that when the flood suddenly came, their family members’ first reaction was to save the elderly by removing them to a safe place. Most of the subjects felt anxious when experiencing floods because of the shock they felt as it was the first time they had to deal with flooding. A small proportion of subjects did not feel anxious because they were used to dealing with floods that often occurred in their area. To ease the anxiety, most subjects prayed and performed religious rituals. When asked about whether or not the subjects had participated in disaster preparedness training, all subjects admitted that they had never participated in such training.

## 4. Discussion

Older people are more vulnerable to the immediate impact of natural disasters and suffer more from injuries and loss of life in them than do younger people. They often have a lack of access to healthy and nutritious food, are susceptible to exposure to infectious and degenerative diseases, and experience stress. Therefore, older people should meet the recommended macronutrient intake. The study objective was to assess food bar supplementation on changes in nutrition and disaster preparedness knowledge, macronutrient intake, and body weight over two weeks. At the time of screening, it was found that variations in the subjects’ nutritional status were malnourished and normal in almost the same proportions, but none were overweight and obese. The proportion of weight gain in the first and second week of the study was the same, both in subjects with low nutritional status and subjects with normal nutrition. However, the proportion of the increase in macronutrient intake in normal nutrition subjects was slightly more significant than in undernutrition subjects. Nutrient absorption in elderly undernutrition is slower than normal elderly nutrition. Malnutrition in older people affects the function and recovery of every organ system in humans: it impairs liver, gut and renal function [25]. The effect of aging on small intestinal motility is lowering the migrating motor complex [26].

The study indicated that broccoli-soybean-mangrove food bar supplementation with nutrition and natural disaster preparedness of older people has a positive effect in improving dietary practices and weight. However, there were some difficulties convincing and motivating the subjects to consume broccoli-soybean-mangrove food bar. Consequently, some subjects did not want to eat daily meals after consuming the food bar even though, we had told them that food bar consumed as a snack, not meals replacement. Among those whose insufficient macronutrient intakes below the Indonesian RDA was relatively low, except fat intake, the broccoli-soybean-mangrove food bar supplementation was successful in achieving macronutrient intakes and weight. Potentially, broccoli-soybean-mangrove food bar has quite high energy content in mangroves (Figure 3), broccoli, and soybean flours as the basic ingredients of food bar. Each 100 g of mangrove fruit flour (Figure 4) contains 88.9 g of carbohydrates and 0.77 g of fat; 100 g grams of broccoli flour contains 45.3 g of carbohydrates and 2.63 g of fat; 100 g of soy flour contains 29.9 g of carbohydrates and 20.6 g of fat [17]. Every 100 g of broccoli-soybean-mangrove food bar contains 492 kcal of energy, 60.4 g of carbohydrates, 3.8 g of protein, and 24.2 g of fat [27].

The increased nutritional knowledge of older people after being given nutrition education in the study is in line with a study in Korea [29]. The nutrition education is effective in improving the nutritional status of older people. Nutrition education can provide adequate knowledge on nutrition and increase the practice of healthy eating in older people [30]. Food acceptance and food preferences of subjects toward the food bars as shown in the hedonic test encouraged food bar consumption behavior and may have increased the energy intake. The increased weight of subject after consuming broccoli-soybean-mangrove food bars in the study is consistent with a study of energy-dense food supplementation in 50 malnourished older people in Canada that shows an increase of 2 kg in the treatment group after 12 weeks. Statistically, the energy, carbohydrates, and fat intakes are significantly different in the treatment group. This significant difference may have played a role in the increase in the weight of the subjects [31]. A dietary supplementation study for 21 undernourished older people in Glasgow, UK, showed weight gain of 2 kg and energy intake during the 3 months of the intervention. Oral nutrition supplementation (ONS) combined with nutrition education is the most effective intervention to increase weight of malnourished older adult.

The proportion of malnourished subjects experienced weight gain was a slight greater than normal nutrition subjects in the first and second week of the study. However, the proportion of normal nutritional status subjects that had an increased intake of macronutrients were higher than malnourished subjects. The proportion of malnourished subjects was almost balanced with the normal nutritional status subjects. Ageing process in humans is accompanied by impaired food absorption, digestion, and metabolism. Some physiological changes in older people, such as reduced saliva production, appetite, and taste will reduce macronutrient intakes. Consequently, many older people have an increased risk for malnutrition compared with other adult populations [32]. The description of the risk of malnutrition in the older people was assessed using the Mini-Nutritional Assessment). The majority of subjects had a risk of malnutrition with a score of <23.5. Although, the mean nutritional status of the subjects was classified as normal (BMI ≥ 18.5 kg/m^2^) [33], this did not necessarily indicate an MNA score greater than 23.5. The findings of this study reveals an inconsistency between the nutritional status proportion of older people as measured by MNA and BMI. This is supported by a study on older people in Gowa which illustrated that the proportion of normal nutritional status in older people using the BMI indicator (67.5%) did not equal good nutritional status using the MNA indicator (42.5%) [34]. Nutritional status determination using the MNA aims to identify whether a person is at risk of malnutrition or not so that early nutritional interventions can be given without having to involve a special team of nutritionists. The MNA is influenced by the environment/place of residence, medical therapy, the presence of pressure wounds, daily frequency of eating, type of protein intake, vegetables or fruit consumption, fluid intake, eating habits, and the perception of the older people about their nutritional status and health [35,36,37].

The functional status or level of independence of the older people was assessed using BADL and IADL tools in this study. Almost all the subjects were independent and able to do their basic daily activities such as controlling urination and bowel stimulation, cleaning themselves, using the toilet, eating and drinking, moving to and from bed in a wheelchair, walking, getting dressed, going up and down the stairs, and taking a shower. In addition, almost all the subjects were able to carry out physical activities outside home as well as at home such as using the telephone, traveling to a place, shopping, preparing food, doing household works, washing clothes, arranging medicines, and managing finances. The results of this study demonstrate that older people with normal nutritional status had a good level of independence inside and outside the home. This is in contrast with the findings of a study on older adults with normal nutrition and over-nutrition in hospitals, showing that overweight older people experienced IADL and BADL limitations in performing their physical activities [38]. This difference may be due to the setting of the second study, which was in a hospital. Furthermore, another study in Taiwan on older people undernourished or at risk of undernutrition also stated that undernutrition drastically increases dependency in BADL and IADL [39]. For cognitive impairment or risk of dementia, the MMSE (Mini-Mental State Examination) was used in this study. One third of total subjects experienced probable cognitive impairment and most of them having normal nutritional status. This finding is contrary to two studies on older people that proved a correlation between undernourished older people and cognitive impairment [40,41].

## 5. Conclusions

The findings of study showed that broccoli-soybean-mangrove food bar supplementation affects weight gain, which was reflected in the significant difference of weight, energy intake, and increased knowledge about a balanced diet among older people at the end of the study. Malnourished subjects had a higher proportion of weight gain than normal nutritional status subjects in the first and second week. The provision of oral nutritional supplementation (ONS) in the form of food bars needs to be accompanied by nutritional education activities to increase adherence to supplementation consumption. Broccoli-soybean-mangrove food bars are an emergency food product containing complete macro-nutrition so that they can be a source of nutrition for older people following natural disasters for 15 days from the start of the evacuation. Raw materials of mangrove, broccoli, and soybean are the local food ingredients for Indonesians people.

However, there were some limitations as the study only assessed older people with two types of nutritional status (underweight and normal nutrition), the intervention time was quite short with a small sample size, and there was no control group. Furthermore, the study also did not measure albumin, lymphocyte, and hemoglobin levels that can be used as indicators of immunity after food bar supplementation. It is recommended that similar research be carried out among overweight and obese older people following a natural disaster. To overcome iron deficiency and to increase immunity of undernourished older people, iron fortification and prebiotics can be added to the food bar for a higher number of older people over a longer period, i.e., 3 months. The objective of the study would be to achieve maximum results in demonstrating a significant change in weight, BMI, hemoglobin level, and lymphocyte level. Research with a control group given a placebo food bar for three months with a large number of subjects is also needed in the future.

## Figures and Tables

**Figure 1 ijerph-18-03686-f001:**
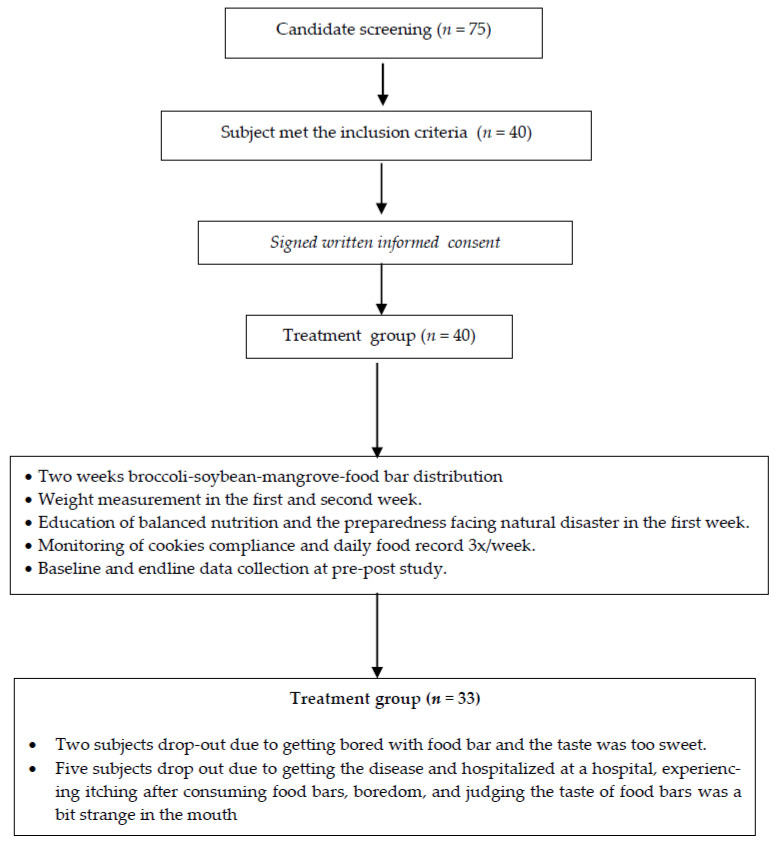
The study scheme.

**Figure 2 ijerph-18-03686-f002:**
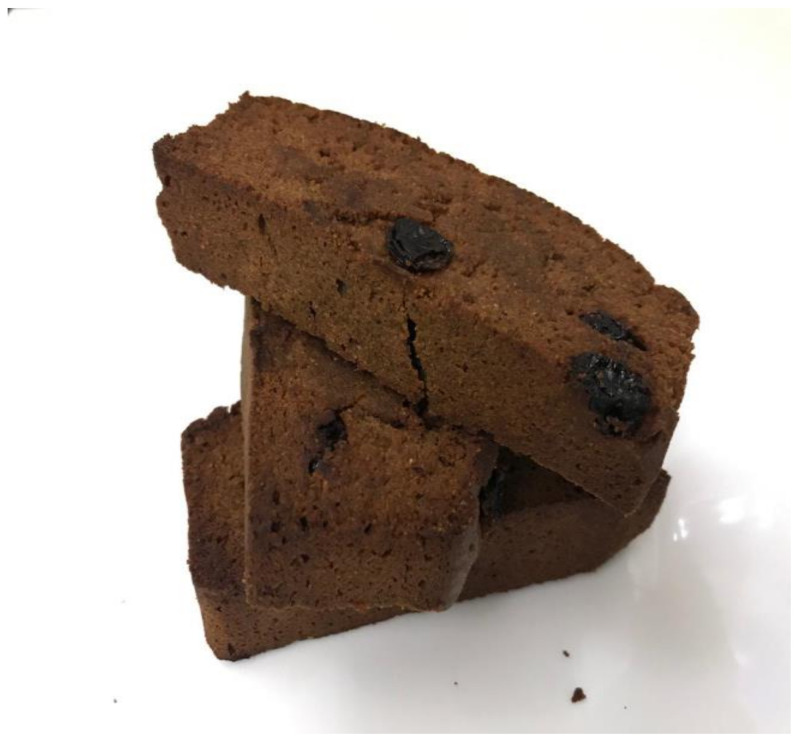
Broccoli-soybean-mangrove food bar (*Bruguiera gymnorrhiza*).

**Figure 3 ijerph-18-03686-f003:**
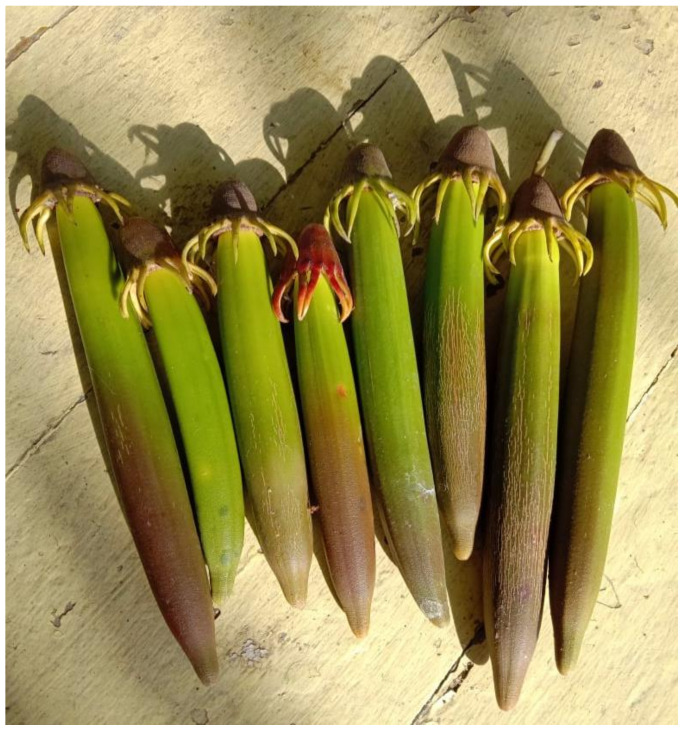
Mangrove/*lindur* fruit.

**Figure 4 ijerph-18-03686-f004:**
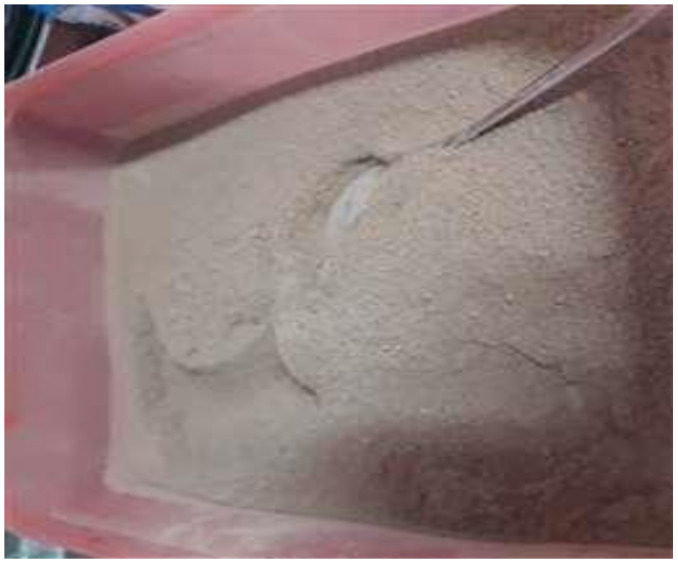
Mangrove/*lindur* fruit flour.

**Table 1 ijerph-18-03686-t001:** Acceptability test.

Level of Likeness	Broccoli Soybean Mangrove Food Bar							
	Aroma		Texture		Taste		Color	
	*n*	%	*n*	%	*n*	%	*n*	%
Dislike extremely	2	7.1	0	0.0	2	7.1	0	0.0
Dislike	11	39.3	1	3.6	5	17.9	4	14.3
Like	13	46.4	25	89.3	18	64.3	20	71.4
Like extremely	1	3.6	2	7.1	2	7.1	3	10.7

**Table 2 ijerph-18-03686-t002:** The hedonic quality test.

Attribute	Broccoli Soybean Mangrove Food Bar	
	*n*	%
Aroma:		
It was very bad	4	14.3
It was a little nice	11	39.3
Good	11	39.3
Very good	2	7.1
Texture:		
It was not very crunchy	2	7.1
It was a bit crunchy	12	42.9
It was crunchy	13	46.4
It was very crunchy	1	3.6
Taste:		
It was not very sweet	2	7.1
It was a little sweet	8	28.6
It was sweet	16	57.1
It was very sweet	2	7.1
Color:		
It was not attractive	2	7.1
It was a bit attractive	5	17.9
It was attractive	20	71.4
It was very attractive	1	3.6

**Table 3 ijerph-18-03686-t003:** Characteristic of socio-demography, nutritional status, cognitive performance, and level of independence profiles.

Indicator	*n*	%	Mean ± SD
Sex			
Male	6	18.2	
Female	27	81.8	
Marital status			
Married	13	39.4	
Widow/widower	20	60.6	
Age (y.o)			70.4 ± 7.8
60–70	14	42.4	
71–80	16	48.5	
≥81	3	9.1	
Final education level			
Low	29	87.9	
Moderate	4	12.1	
Staying at home			
Alone	3	9.1	
Husband/wife/4/12.1			
Children/grandchild/son-in-law	23	69.7	
Another family member	3	9.1	
Body Mass Index/BMI (kg/m^2^)			19.6 ± 3.5
Underweight (<18.5 kg/m^2^)	15	45.5	
Normal (18.5–24.9 kg/m^2^)	18	54.5	
Overweight (25.0–29.9 kg/m^2^)	0	0.0	
Obesity (≥30.0 kg/m^2^)	0	0.0	
MUAC (Mid Upper Arm Circumference) (cm)			22.5 ± 2.8
MNA (Mini Nutritional Assessment)			21.2 ± 3.9
At risk malnourished	25	75.8	
Normal	8	24.2	
MMSE (Mini Mental State Examination)			21.1 ± 6.4
Cognitive impairment	8	24.2	
Probably cognitive impairment	11	33.3	
Normal	14	42.5	
BADL (Basic Activity Daily Living)			19.8 ± 0.9
Light dependence	3	9.1	
Independently	30	90.9	
IADL (Instrumental Activity Daily Living)			5.1 ± 2.3
Need help all the time	2	6.1	
Need help now and then	3	9.1	
Independently	28	84.8	

**Table 4 ijerph-18-03686-t004:** Change of weight at pre-post study.

Nutritional Status at Pre-Study	In the First Week						In the Second Week					
	Loose (Person)		Stay (Person)		Gain (Person)		Loose (Person)		Stay (Person)		Gain (Person)	
	*n*	%	*n*	%	*n*	%	*n*	%	*n*	%	*n*	%
Underweight	6	50.0	0	0.0	9	50.0	3	33.3	2	50.0	10	50.0
Normal	6	50.0	3	100.0	9	50.0	6	66.7	2	50.0	10	50.0
Total	12	100.0	3	100.0	18	100.0	9	100.0	4	100.0	20	100.0
Mean (kg)	−0.30		0.0		0.34		−0.29		0.0		0.50	
Minimum (kg)	−0.80		0.0		0.10		−1.20		0.0		0.10	
Maximum (kg)	−0.10		0.0		0.90		−0.10		0.0		1.20	

**Table 5 ijerph-18-03686-t005:** Change of macronutrient intakes at pre-post study.

Nutritional Status at Pre-Study	Energy Intake				Carbohydrate Intake				Protein Intake				Fat Intake			
	(kcal)				(gr)				(gr)				(gr)			
	Loose		Gain		Loose		Gain		Loose		Gain		Loose		Gain	
	*n*	%	*n*	%	*n*	%	*n*	%	*n*	%	*n*	%	*n*	%	*n*	%
Underweight	5	71.4	10	38.5	4	66.7	11	40.0	8	66.7	7	33.3	5	62.5	10	40.0
Normal	2	28.6	16	61.5	2	33.3	16	59.3	4	33.3	14	66.7	3	37.5	15	60.0
Total	7	100.0	26	100.0	6	100.0	27	100.0	12	100.0	11	100.0	8	100.0	25	100.0

**Table 6 ijerph-18-03686-t006:** Anthropometry, macronutrient intakes, and nutrition and natural disaster preparedness of older people changes.

Measurement	Mean	SD	95% CI	*p*
Body weight (baseline vs. week 1)	41.6	8.3	26.8–64.2	>0.01
Body weight (week 1 vs. week II)	41.7	8.2	26.9–64.2	<0.01
Body weight (baseline vs. weeks I)	41.8	8.2	26.7–64.4	<0.01
Energy (kcal)	2291.9	210.0	217.4–366.4	<0.01
Protein (g)	6.1	13.3	1.33–10.8	<0.01
Fat (g)	15.6	14.6	10.4–20.8	<0.01
Carbohydrate (g)	31.1	46.9	14.5–47.8	<0.01
Nutrition and natural disaster preparedness knowledge of older people (baseline-week II)	11.8	21.7	4.1–19.5	<0.01

**Table 7 ijerph-18-03686-t007:** Comparison of mean macronutrient intake with percentage of recommended dietary allowance (RDA) during study.

	Mean (SD) Macronutrient Intake								Mean Macronutrient Intakes as Percentage of RDA			
Nutrient	Baseline				At the Second Week				Baseline		At the Second Week	
	Mean	DS	Mean	DS	Mean	DS	Mean	DS	Male	Female	Male	Female
Energy (kcal)	828.60	317.56	52.67	15.62	1271.90	296.04	69.33	14.19	46.03	52.67	70.66	69.33
Protein (g)	29.90	11.40	47.81	20.25	37.68	10.13	57.62	17.73	46.74	47.81	58.88	57.62
Fat (g)	27.95	9.39	63.47	28.66	45.05	10.57	97.32	21.67	55.90	63.47	90.10	97.32
Carbohydrate (g)	129.67	80.90	52.77	19.52	180.68	50.52	64.38	11.68	47.15	52.77	65.70	64.38

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
