# Peer review of "Broccoli-Soybean-Mangrove Food Bar as an Emergency Food for Older People during Natural Disaster"

_ijerph, 2021, doi:10.3390/ijerph18073686_

Round 1
Reviewer 1 Report
To my opinion, the data presented are interesting but not appropriately presented.
In the introduction has to be presented the objective of the study, but not the composition of the snack bar what has to be more common to the part of materials and methods.
The study design is described inaccurately.
Why organoleptic test was performed on 27 older people who were not included in the experiment?
Why figure 1 is in the part of the results?
How was prepared broccoli-soybean-mangrove-food bar?
What the connection of Hedonic and Hedonic Quality Test of the bar with the overall study?
Conclusions have to be revised.
Reviewer 2 Report
The manuscript shows an important problem of malnutrition of the elderly. The research was conducted on a small group (33 persons), but provides the basis for further studies.
The methodology of making bars must be supplemented. The different composition of the bars was described:
L82 – 86: “A study on the development of an emergency food product in the form of a food bar that is made from a mixture of mangrove fruit or lindur fruit flour (Bruguiera gymnorrhiza), broccoli flour, and soy flour…”
L104: “The treatment group was given broccoli-soybean-mangrove-food-bar during two weeks”
The manuscript should be improved. The questions were asked in the manuscript. Please see attached file.

Reviewer 3 Report
This an is an interesting study of a suplement of a 50 g food bar given daily to improve nutrition modestly.
Do the investigators think their estimate of number of subjects required was too low.
Their population was obese but they can become quite malnourished with muscle wasting.
Do the investigators think they can identity he see group. They allude to this in the final paragraph of the Discussion. Do they want to enlarge on this idea.
I found the Conclusions weak, disjoint, and just platitudes. They could hav eremarked on the elderly obese with malnutrition.
Its seem to me many of the ideas about current malnutrition come from studies in Africa of 1950s and 1960s and need to revised and made contemporaneous.
At least this study is of th elderly which is more pertinent of Today.
Round 2
Reviewer 1 Report
Dear Author,
Thank You for the improvements made, but some additional observations:
Line 146-147, why if it was used the same baking temperature it's separated in two times?
Figure 1 not only mentioned but all have to be moved in the part of the methodology.
Author Response
Dear Reviewer 1,
Please find the revised my manuscript which has been revised according to your inputs as follows:
- Line 146-147 --------> the explanation put in the line 160-162 (green color).
- Figure 1 has been moved including the explanation of subject recruitment in the line 123-133 (green color).
Thank you for these valuable inputs.
Regards
Fatmah